# Impact of the COVID-19 Pandemic on Adolescents’ Sexual and Reproductive Health in Low- and Middle-Income Countries

**DOI:** 10.3390/ijerph182413221

**Published:** 2021-12-15

**Authors:** Salima Meherali, Bisi Adewale, Sonam Ali, Megan Kennedy, Bukola (Oladunni) Salami, Solina Richter, Phil E. Okeke-Ihejirika, Parveen Ali, Kênia Lara da Silva, Samuel Adjorlolo, Lydia Aziato, Stephen Owusu Kwankye, Zohra Lassi

**Affiliations:** 1Faculty of Nursing, University of Alberta, Edmonton, AB T6G 1C9, Canada; adewale@ualberta.ca (B.A.); ssali2@ualberta.ca (S.A.); mrkenned@ualberta.ca (M.K.); bukola.salami@ualberta.ca (B.S.); 2College of Nursing, University of Saskatchewan, Saskatoon, SK S7N 5E5, Canada; solina.richter@usask.ca; 3Women’s & Gender Studies, University of Alberta, Edmonton, AB T6G 1C9, Canada; pokeke@ualberta.ca; 4Division of Nursing and Midwifery, Health Sciences School, University of Sheffield, Sheffield S10 2LA, UK; parveen.ali@sheffield.ac.uk; 5Programa de Pós-Graduação em Enfermagem, Federal University of Minas Gerais, Belo Horizonte 30130-100, MG, Brazil; kenialara17@gmail.com; 6Department of Mental Health Nursing, University of Ghana, Legon, Accra P.O. Box LG 43, Ghana; sadjorlolo@ug.edu.gh; 7School of Nursing and Midwifery, University of Ghana, Legon, Accra P.O. Box LG 43, Ghana; laziato@ug.edu.gh; 8College of Humanities, Regional Institute for Population Studies (RIPS), University of Ghana, Legon, Accra P.O. Box LG 43, Ghana; skwankye@ug.edu.gh; 9Robinson Research Institute, University of Adelaide, Adelaide, SA 5005, Australia; zohra.lassi@adelaide.edu.au

**Keywords:** COVID-19, adolescents, sexual health, reproductive health, health services, access, interventions, low- and middle-income countries

## Abstract

Adolescents living in low- and middle-income countries (LMICs) are struggling with accessing sexual and reproductive health (SRH) services, and COVID-19 has escalated the problem. The purpose of this review was to identify and assess the existing literature on the impact of the pandemic on SRH needs and access to services by adolescents in LMICs. A scoping review was conducted to collate findings on the topic. Searches were performed on eight databases. Data were extracted and categorized into various themes. After removing duplicates and performing a full-text reading of all articles, nine articles were included in our review. Our findings generated several themes related to adolescents’ sexual and reproductive health during the COVID-19 pandemic. These include (1) limited access to sexual and reproductive health services, (2) school closure and increased rate of early marriages, (3) sexual or intimate partner violence during COVID-19, (4) disruption in maternity care, (5) adolescents’ involvement in risky or exploitative work, (6) intervention to improve sexual and reproductive health services during COVID-19, and (7) policy development related to adolescent sexual and reproductive health. Several recommendations were made on policies—for instance, the use of telemedicine and community-based programs as a way to deliver SRH services to adolescents during and after a pandemic.

## 1. Introduction

As the fastest-moving global public health crisis during this century, the COVID-19 pandemic is causing significant mortality and morbidity and creating daunting health and socioeconomic challenges. The restrictive measures that have been put in place by many countries to contain the spread of the virus may negatively affect access to essential sexual and reproductive health (SRH) services, particularly by adolescents living in low- and middle-income countries (LMICs) and fragile settings [1]. Young people are especially affected by the closure of social spaces, including schools, community centers, and health clinics, where many of them receive comprehensive education on SRH and services. Past epidemic outbreaks have shown that disruptions in education are extremely harmful to young people, especially girls, in terms of not only lost earnings and education but also increased vulnerability to gender-based violence, early marriages, unintended pregnancy, and female genital mutilation [2].

Globally, approximately 1.8 billion people are between the age of 10 and 19 years, and 90% of them live in low- and middle-income countries (LMICs). Low- and middle-income-earning countries are countries that are considered either as low-income-earning countries or both low and high middle-income-earning countries. According to the World Bank, low-income economies are countries with a Gross National Income (GNI) per capita of USD 1025 or less, and middle-income economies (low and high middle-income countries) are those with a GNI per capita between USD 1026 and USD 12,375, using the World Bank Atlas method [3]. Thus, all countries with a GNI per capita of USD 1025 or less to USD 12,375 using the World Bank Atlas method can be classified as low- and middle-income-earning countries. Many adolescents from LMICs, particularly girls, are vulnerable to poor SRH and experience early and unintended pregnancy, unsafe abortions, sexual violence, and sexually transmitted infections. These adolescents already faced significant barriers in accessing essential SRH information and services before the COVID-19 crisis [4]. Lockdowns and the diversion of medical resources have led millions of women and girls to be forced to carry unwanted pregnancies or risk unsafe abortions [5]. In addition, a low level of access to SRH information and services has resulted in increased teenage pregnancies throughout the world but more so in many LMICs [6]. Potential disruption of the supply and the production of contraceptives also leads to teenage pregnancies in LMICs. In addition, the redeployment and unavailability of medical staff and breakdown of SRH services have also affected the availability and accessibility of family planning services [7,8]. According to the Guttmacher Institute (2020), disruptions in access to contraceptive services and disruptions in prevention programs may lead to 13 million more child marriages over the next decade in LMICs [2,9].

The pandemic has caused further strain on health systems, specifically in LMICs, and has likely impacted adolescents’ SRH and access to SRH services. However, there is no comprehensive and integrated literature on the impact of the pandemic on SRH access and use among adolescents in LMICs. Our review has addressed this gap through an exploration of the SRH priorities and access to SRH services by adolescents in LMICs during the pandemic. The purpose of this scoping review was to identify and summarize the already existing research and literature regarding the impact of the pandemic on SRH needs and access to services by adolescents in LMICs. Specifically, this study aimed to identify research gaps in the existing literature, as well as to summarize and disseminate research findings for clinicians, service providers, and policymakers. The research questions that guided the study were:What is known from the literature about the impact of COVID-19 on SRH of adolescents in LMICs?What interventions have been designed and implemented to promote SRH and access to SRH services for adolescents in LMICs during the COVID-19 pandemic?How can future studies overcome research gaps and inform programs and policies that meet the SRH care needs of adolescents in the context of the pandemic?

## 2. Methods

To explore the impact of the COVID-19 pandemic on the SRH of adolescents residing in LMICs and the types of interventions designed to improve access to SRH services, we used a scoping review methodology as guided by Arksey and O’Malley [10] and Levac et al. [11]. Scoping reviews are well-suited for areas where there is limited knowledge or research to identify the extent, range, and nature of research activity in a given area, as well as to identify research gaps in the existing literature. This paper adheres to the Preferred Reporting Items for Systematic Reviews and Meta-Analyses (PRISMA) extension for scoping reviews [12]. Typically, scoping reviews do not require a quality appraisal of the articles reviewed. Hence, we did not conduct a quality appraisal of the included articles as our aim was to map all research activities in this field. The review has been registered in the OSF database with DOI number: 10.17605/OSF.IO/AD752.

### 2.1. Data Sources and Search Strategy

A systematic literature search was conducted by an experienced health sciences librarian (MK) to identify all relevant published studies. Searches were performed in the following databases: Medline, EMBASE, PsycINFO, HealthSTAR, Sociological Abstracts via ProQuest, Cumulative Index for Nursing and Allied Health Literature (CINAHL) via EBSCOhost, Scopus via Elsevier, and the Cochrane Library via Wiley. These databases were searched using a combination of natural language vocabulary and controlled terms (subject headings) wherever they were available. Natural language terms were derived from three main concepts: (1) sexual reproductive health issues including, sex, sexuality, pregnancy/family planning, sexually transmitted infections, and others; (2) LMICs as defined by the World Bank; (3) adolescents. A few of the keywords used for the search in the electronic databases included adolescents, young adults, sexual health, reproductive health, reproductive rights, LMICs, COVID-19, pandemic, access to health services, teenage pregnancy, contraceptive, sexuality education, gender-based violation, unintended pregnancies, and female genital mutilation. In order to increase the search sensitivity, publication date, language, and study type restrictions were not applied. Other search methods were employed in order to retrieve additional evidence. These methods included hand searches of the reference lists and forward citation searches, using Scopus, of papers to be synthesized in the scoping review. Finally, the team reviewed the grey literature on this topic. All searches were performed in February 2021 (see Appendix A for full search strategies by database).

### 2.2. Study Selection

Articles were included if the study examined adolescents’ needs related to SRH, the impact of the COVID-19 pandemic on SRH, access to SRH education and services, and needs related to approaches to SRH education and services. We included studies conducted on adolescents in LMICs (10 to 19 years). We included studies that involved broader age groups provided that they had subgroup data for 10- to 19-year-olds. Articles published in English from March 2020 to date were included. All primary research studies irrespective of study design were considered. Using Covidence, a web-based tool that helps to identify studies and involves data extraction processes [13], two reviewers (BA and SA) independently screened all potential articles. In case of disagreement, both reviewers read the paper and discussed it until they reached a consensus. The reviewers (BA and SA) independently screened all eligible full-text articles and included papers in this review that satisfied all of the inclusion criteria. Articles that only had abstracts and research protocols were excluded. Articles that did not focus on adolescents in LMICs, access to SRH services during the pandemic, interventions implemented to promote SRH, and access to SRH services for adolescents during the pandemic were excluded. In addition, reports that did not focus on SRH health experiences and/or outcomes of adolescents during the pandemic in LMICs were excluded from this review. Articles that focused on adolescent mental health were also excluded from this review.

### 2.3. Data Extraction

We extracted relevant data from each study, including the year, study design, setting, the target population, and the outcomes that the researchers measured (see Table 1). One reviewer (BA) used a form that the research team developed to extract data. A second reviewer (SA) verified all of the data extraction and checked for accuracy and completeness. We resolved disagreements through discussion.

## 3. Results

The initial search retrieved a total of 261 articles. After removing duplicates and articles in other languages and reviewing abstracts with respect to the inclusion criteria, a total of 25 studies were considered relevant. Following a full-text review and consultation among the reviewers, nine articles were included in the final review and analysis using the PRISMA diagram (Figure 1) [23]. Findings from each article were summarized in a table format and systematic analysis was performed to extract major themes. A descriptive synthesis table was formulated containing the textual descriptions of all the findings. A detailed analysis was performed to evaluate the impact of the COVID-19 pandemic on access to SRH services among adolescents in LMICs and the effectiveness of interventions designed to promote SRH and access to SRH services among this population during and after the pandemic. A summary of all nine selected papers, including author and place of the study, study design, sample size, intervention, and outcomes, is provided in Table 1. Four of the studies included were qualitative [14,15,16,17], two were quantitative [18,19], one was a policy analysis [20], one was a program follow-up [21], and one was a technical report [22].

A systematic analysis of the findings was conducted to identify relevant themes. Adolescents faced various SRH vulnerabilities during COVID-19, which can be categorized into several themes, including limited access to sexual and reproductive health services, school closure and increased rate of early marriages, sexual or intimate partner violence during COVID-19, disruption in maternity care, and adolescents’ involvement in risky or exploitative work. The review also generated other themes, such as interventions to improve sexual and reproductive health services, and policy development related to adolescent sexual and reproductive health during COVID-19. Findings of these issues and interventions are discussed in the following section in detail.

### 3.1. Limited Access to Sexual and Reproductive Health Services during COVID-19

Limited access to youth-friendly healthcare services was reported in various countries during the COVID-19 pandemic. In Kenya, some adolescents (17%) complained of not being able to go to their regular clinic for medical care, and others (3%) reported not being able to get their medication refills. More specifically, 2% of adolescents aged 15 to 19 years were not able to access medication refills, whereas 8% of adolescents reported missing antiretroviral therapy (ARVs) for 2 or more days in a row in the last 30 days [15].

In addition, many older girls reported ignoring and not being able to access the needed SRH services during the pandemic, such as modern contraceptive and family planning services [18]. Adolescents in Cote d’Ivoire also reported difficulty accessing contraceptives due to disruptions to clinic hours and outreach, which was attributed to an increased rate of adolescent pregnancies [14]. Accessing and purchasing menstrual hygiene products was another issue reported by adolescents amid COVID-19. In Lebanon, access to supplies for menstrual management was also a concern among refugee girls, who either resorted to borrowing menstrual products from local stores or using pieces of cloth at home [14].

A study conducted in Syria reported that teenage mothers expressed financial distress due to the pandemic, which affected their ability to buy sanitary pads. Alternatively, young girls resorted to using whatever material or cloth was available at home as they had no money to buy sanitary pads [14]. Similar experiences were reported in another study conducted in Kenya. Approximately 53% of girls between 15 and 19 years who had begun menstruating reported difficulty obtaining their preferred menstrual hygiene management products, since the majority of them or their families no longer had enough money to purchase them [18].

In contrast, despite the lockdown situation in a country such as South Africa, it was found that adolescents’ visits for perinatal care and family planning remained reasonably constant or modestly increased during the pandemic lockdown [19]. An estimated 20% increase in clinic visits for HIV-related care and treatment immediately after the lockdown was reported, and it is suspected that this might be because of an urgency to collect medications before there was an anticipated interruption in clinic access or medication availability and/or national programmatic efforts to accelerate transitions to a new first-line regimen [19]. Similarly, Siedner et al. [19] observed resilience in family planning visits by adolescents over the observation period, which increased from 7.3 visits/clinic/day in the pre-implementation period to 7.8 visits/clinic/day after the transition to level 5 lockdown. The level 5 lockdown in this study was the period of lockdown between 28 March 2020 through to 30 April 2020 [19].

### 3.2. School Closure and Increased Rates of Early Marriages

A study conducted in three different countries, including Ethiopia, Cote d’Ivoire, and Lebanon, reported mixed findings on the risks of child marriage amid the COVID-19 pandemic [14]. In Lebanon, two of the adolescent Palestinian girls became engaged during the lockdown, while other adolescent girls mentioned postponing their marriage due to economic crisis, resulting in lower marriage rates [14]. Similar findings were reported by both Palestinian and Lebanese boys, who perceived the pandemic and resulting economic challenges as a major barrier to their marriage. On contrary, the marriage rate was perceived to be higher among Syrian girls as marriage to a Syrian girl in comparison to a Lebanese girl is considered cheaper as they carry a high dowry with them upon marriage [14]. Moreover, both adolescent girls and boys in Ethiopia were pressurized to get married as school closure coincided with the traditional wedding season in three of the six communities where the study took place. Even daughters in their early adolescence were forced to get married because of the limited presence of local authority officials and schoolteachers, who otherwise would cause hindrances in early child marriages. In addition, adolescent boys who previously attended secondary schools were also under pressure to get married during the stay-at-home restrictions [14].

### 3.3. Sexual or Intimate Partner Violence during COVID-19

The frequency of intimate partner violence rose amid this pandemic. Adolescents reported being at high risk of experiencing intra-household violence and, in the case of Lebanon and, to some extent, Côte d’Ivoire, girls reported being at a heightened risk for community-level violence [14]. In addition, married Syrian girls also reported increased tensions with their husbands, in-laws, and neighbors, along with reports of intimate partner violence [14]. Married girls in Ethiopia also became victims of intimate partner violence due to financial hardships and lockdown situations [14]. The lockdown caused male partners to easily become frustrated and violent, as reported in an Ethiopian study. In this study, a female partner reported that her husband became easily irritated when she asked for help in taking care of their children while she took care of other household chores and hit her with a stick on her head [14]. On the contrary, in another study conducted in urban and rural regions in Kenya, the majority (45%) of adolescent girls reported less violence at the beginning of the COVID-19, as compared to only 6% of adolescents who reported more violence, including physical, emotional, and sexual violence, and other forms of violence [18].

### 3.4. Disruption in Maternity Care

In Afar, a town in Ethiopia, married girls were concerned about infection transmission and remained fearful of seeking antenatal care, due to which they opted for home-based births, resulting in several maternal deaths [14]. One other study [17] included in this review showed that access to maternity care for pregnant women was interrupted during the COVID-19 lockdown. Muhaidat et al. [17] conducted a study in Jordan that reported an increase in the number of women unable to access antenatal care during the COVID-19 curfew. In this study, only 4% (38/944) of study participants reported not having access to antenatal care prior to the lockdown. However, the study found that there was an increase in the number (59.5%) of participants not having access to an antenatal clinic during the lockdown [17]. This indicated that more than half of the participants did not have access to antenatal care during the lockdown. The remaining 40.5% (382/944) had access to antenatal care as previously before the lockdown (3.81%), via telephone (28.07%), through medical emergency departments (2.97%), and through the use of a permit to commute (5.61%) during the lockdown [17]. Further, the study by Muhaidat et al. [17] showed that the majority of women who experienced complications in pregnancy were the ones who did not have access to antennal care.

### 3.5. Adolescents’ Involvement in Risky or Exploitative Work

The pandemic caused great financial challenges to families and communities during the lockdown. As a result, youth were compelled to engage in or consider risky or exploitative work [14]. In Ethiopia, adolescent girls in towns and cities were noted to be vulnerable to sexual harassment and abuse while working. This was largely because of the closure of bars, which led people to turn to locally brewed alcohol and traditional alcohol houses. Banati, Jones, and Youssef [14] found that adolescent girls who helped their mothers in making and selling ‘tella’ (a local alcoholic drink) were exposed to unwanted sexual attention and sexual abuse.

### 3.6. Interventions to Improve Sexual and Reproductive Health Services during COVID-19

Interventions to make SRH service available and accessible to adolescents were proposed during the COVID-19 pandemic lockdown. The UNFPA [22] was one of the international organizations that outlined interventions that could be used by countries to make SRH services accessible and available through local adolescent-friendly mass and digital media platforms. Information regarding comprehensive sexual education, and how this can be delivered outside of schools during the COVID-19 era using media platforms, was highlighted [22]. Similarly, the use of media campaigns to help reduce the number of unwanted pregnancies amongst adolescent girls during COVID-19 was acknowledged by Eboi and Awan Ismail [16] in a Kenyan study. Eboi and Awan Ismail [16] acknowledged potential challenges with using mass media as a platform to campaign against unwanted pregnancies amongst adolescent girls in Kenya. These challenges included cultural beliefs and taboo around this topic, a lack of technical skills among implementers, a lack of resources, and social media mistrust. Nevertheless, their study found that these challenges could be resolved through the involvement of stakeholders in the creation of messages, lobbying for funds, and engaging in training on technology.

Another intervention was the involvement of healthcare professionals to ensure that adolescents had access to age-specific, accurate, and up-to-date information on SRH verbally through the media and pamphlets. This information included access to information on contraceptives, comprehensive care, antenatal, intrapartum, and postnatal care, testing and care for HIV and other STIs, and menstrual health available to adolescents during the lockdown [22]. Furthermore, on the accessibility of contraceptives, it was suggested by UNFPA [22] that restrictions such as age, marital status, and consent from parents or spouses for contraceptive use needed to be reviewed during this era. The provision of contraceptives to last for longer periods or months, as well as the use of alternative delivery modalities for contraceptives, such as pharmacies and community-based delivery, was another intervention suggested [22].

Additionally, telemedicine was encouraged as an intervention to provide services for safe and easy access to medical abortions, counselling, and screening to adolescents [22]. Moreover, adolescent-friendly phone lines that would provide advice on the side effects of contraception, contraceptive self-use, and available contraceptive options could be a beneficial intervention. Furthermore, other suggested the testing of adolescents for HIV and other STIs at home, and sending results through messaging, ensuring privacy and confidentiality [22]. Similarly, Dourado et al. [21] mentioned the use of e-services and home delivery services as a strategy for adolescents’ access to HIV Self-Test (HIVST) kits, distribution of condoms, HIV pre-exposure prophylaxis medications (PrEP), lubricants, and douches in Brazil during the lockdown. These interventions were meant to reduce the need for personal visits to the clinics during this period and encourage adolescents to patronize these services [22]. The distribution of menstrual supplies to adolescent girls during this era was to be considered by communities and governments in various countries. This would aid in limited movement and help adolescent girls to ensure proper menstrual hygiene during the stay-at-home period [22].

Dourado et al. [22] assessed the use of the digital platform as an intervention for adolescents who wanted to socialize during the lockdown and also wanted sexual partners. This intervention posed a significant effect on adolescents who patronized this service during the COVID-19 lockdown in Brazil. Further, the intervention by Dourado et al. [21] used social media platforms and radio programs to talk about sex and hormonal therapies focusing on adolescents. These talks were used to identify adolescents for the PrEP, after which follow-up provision of SRH services was done through telehealth (using videos or voice connection) and in-person visits by appointment [21]. Services provided included rapid HIV serum testing and counselling on the prevention of HIV, and encouraged the use of the HIVST [21].

### 3.7. Policy Development Related to Adolescent Sexual and Reproductive Health

As restrictions were imposed due to COVID-19 in most parts of the world, some countries and international organizations outlined the development of policies on adolescent SRH. Two of the articles [14,20] and one grey literature article [22] included in this systematic review highlighted the need for policy developments for adolescent SRH. These policies were aimed at supporting the adolescents during the COVID-19 lockdown in some countries. Herran and Palacios [20] mentioned the need for policies that would concentrate on the social and cultural barriers and economic inequalities, with the aim of lessening adolescent pregnancy during the COVID-19 era. This approach by Herran and Palacios [20] was based on previous situations similar to the COVID-19 pandemic, which contributed to an increase in the number of adolescent pregnancies. Similarly, Banati, Jones, and Youssef [14], in their study, mentioned the need for community-based and culturally focused models directed towards serving the population in the locality.

As reported by Herran and Palacios [20], communities with limited access to job opportunities, and with existing social isolation, could negatively explore early parenthood as an approach to heightening their social status. Therefore, the need for career counselling and job creation opportunities for disadvantaged adolescent boys and girls in such communities may address this issue. Additionally, the creation of a virtual platform that would offer the chance for the early delivery of psychosocial services to adolescents was recommended by Banati, Jones, and Youssef [14], and the UNFPA [22] highlighted the need for countries to develop policies that would aid access to SRH services to adolescents during COVID-19. Policy recommendations for countries by the UNFPA [22] included the use of available media platforms to provide information to adolescents on access to (a) sexuality education; (b) contraceptive counselling and services; (c) comprehensive abortion care; (d) antenatal, intrapartum, and postnatal care; (e) prevention of HIV and other STIs; (f) prevention, care, and response to sexual and gender-based violence; g) prevention of cervical cancer through HPV vaccination; and (h) counselling and services for sexual health and well-being that include the provision of menstrual health information and services. Moreover, policies to empower adolescent girls, reduce their reliance on male partners, and enable them to stay away from unprotected sex were needed to reduce adolescent pregnancy during this era [20]. The UNFPA [22] in their document also further encouraged countries to institutionalize the approaches used during the lockdown period to improve access to quality services after the curfews.

## 4. Discussion

This scoping review synthesized the literature on the impact of the COVID-19 pandemic on the SRH of adolescents living in LMICs. Despite the growing conversation around SRH and the rights of adolescents amid COVID-19 among national and international stages, a dearth of literature remains on this topic specific to adolescents’ needs in LMICs. To the best of our knowledge, this scoping review is the first of its kind that systematically synthesizes the literature on the impact of the pandemic on SRH and the rights of adolescents in LMICs. The findings from our scoping review add to the discussion on the impact of the pandemic on SRH and access to services by establishing the diverse and multi-faceted SRH service needs specific to adolescents in LMICs. Moreover, our findings demonstrate that adolescents would benefit from programs and policies that address the impact of the pandemic on SRH and access to services in LMICs.

### 4.1. Low Socioeconomic Status and Early or Unwanted Pregnancies

The socioeconomic status of the adolescents contributed to their SRH during the lockdown. The current review suggests an increase in early marriages and unwanted pregnancies during the lockdown, because of the closure of schools. Both adolescent girls and boys were forced into early marriages during the lockdown. With adolescents staying at home long-term during this period, the parents of girls with a low socioeconomic background gave them out for marriage. In most of these communities, a dowry is usually paid to the parents of the bride, and the cost varies amongst communities. Some studies included in this review mentioned the closure of schools as a contributing factor to the increase in early marriages. However, the financial stress experienced in most countries due to COVID-19 [24,25,26] could also be a reason for parents giving their daughters off early for marriage. Questions remain about the adolescent boys who were forced to get married. Were their parents getting them brides at this early age, to take advantage of the global financial crises to pay cheaper dowries? Further studies to understand why adolescent boys were also being pressured to get married early are encouraged. The possibility of most adolescents with low socioeconomic status not returning to school after the lockdown is high and must also be explored.

The review also suggests an increase in unprotected sexual activities, and girls entering into an intimate relationship with the opposite sex during the lockdown, leading to more unwanted pregnancies amongst adolescent girls. In some countries, adolescents are sometimes required to obtain permission or endorsement from their parents, or spouses if married, before using a contraceptive [27,28]. This could have been challenging during the lockdown for some adolescents who were away from their parents. SRH service providers must be prepared to ease restrictions, to allow sexually active adolescents to have access to contraceptives in challenging situations [22]. Such actions need to be considered for future pandemics to reduce the inconveniences that adolescents experience in accessing the needed contraceptives during a lockdown. Adolescent girls and boys could have been enrolled in skilled training of their choice during this period for free, or even given jobs just to keep them busy, as suggested by Herran and Palacios [20]. This would have been beneficial to the adolescent, their family, and the nation, instead of the burden of forced early marriages and unwanted pregnancies that the review found.

### 4.2. Impact of Pandemic on Affordability and Accessibility of Menstrual Products

We suggest that access to menstrual products was also halted during the pandemic due to various factors. Mobility restrictions and girls’ confinement to home limited their ability to buy their desired menstrual products and they were forced to use home-based material to manage menstruation. Affordability was another factor that contributed to the poor access to menstrual hygiene products. Our findings coincide with studies conducted in various regions across the globe that also found that girls suffered from poor menstrual hygiene management during COVID-19 due to limited access, disrupted supply to menstrual hygiene products, particularly disposable menstrual hygiene materials that require monthly replenishment with increased demands and rise in the product prices, a less hygienic environment for sanitary hygiene product disposal, reduced availability of clean water, and access to information about menstrual hygiene [29,30].

### 4.3. Domestic/Intimate Partner Violence during the Pandemic

Adolescents also suffered from domestic violence (DV) and intimate partner violence (IPV), which significantly increased amid the pandemic. Our study highlighted several factors that perpetuated the situation, such as stress due to loss of income, economic instability or poverty, and men staying at home and not going to work due to lockdown orders. Similar findings were reported in other studies, where the households that suffered from unemployment and low income were the ones highly affected by intimate partner violence [31,32] These behaviors could also be attributed to the reduced socialization of women, their isolation from support networks outside home, and women’s inability to escape from abusive situations [32], which allowed violent partners to exert their control and power over women and girls.

Similar to LMICs, cases of DV and IPV were also reported in Argentina, the USA, Pakistan, and several other countries due to pandemic-induced financial worries and the poor mental health of either of the partners [31,32,33]. Moreover, stay-at-home orders further exaggerated the situation. Observations from one of the studies conducted in Argentina found a lower prevalence of violence among women whose partners did not have to comply with a stay-at-home order [34].

### 4.4. Limited In-Person Services Experience

The review found that there was limited access to sexual and reproductive health services. Some adolescents were unable to access medical refills for antiretroviral therapies (ARTs) during the lockdown, hence missing doses of their medication for several days. Moreover, adolescents reported not having access to contraceptives and menstrual hygiene products. This was also reported in other studies and was due to the social distancing or lockdown in most countries during the peak of COVID-19 [35,36]. However, there were alternative measures deployed in some countries to reach out to adolescents and youth, to access some sexual and reproductive health services. For example, the distribution of HIV self-testing kits and condoms in homes and communities and the use of media platforms were deployed as strategies to maintain access to SRH services. The media platforms, when used as a means to communicate during the period of social distancing or lockdown, can be beneficial to a large number of audiences [37,38]; however, this can also be a limitation and challenging to some individuals and communities [39]. The involvement of stakeholders to have an approved media campaign on SRH was suggested by [17] during the pandemic.

### 4.5. Role of School in Promoting SRH

The findings of our study suggest that schools have been playing a major role in delaying early marriages and adolescent pregnancies, and the closure of schools during the pandemic has had a dramatic impact on adolescents’ health. Child marriages can result in serious health consequences, including early and high-risk pregnancy, which may increase the risk of maternal and infant morbidity and mortality [40,41]. It can also isolate girls from their family and social network, resulting in dropouts from schools, and exclude them from participating in their communities, which may drastically affect their mental health and overall well-being [42].

However, numerous studies suggest that keeping children in schools is the best way to prevent early marriages [43]. School is a platform to disseminate and generate awareness of sexual and reproductive health among youth, support their physical and psychosocial development, and engage them in extracurricular activities, thereby distancing them from negative behaviors. Our findings suggest that school teachers and management acted as a protective factor, who supported adolescents’ education and prevented early child marriages, but, due to the lockdown, teachers remained away from schools, consequently increasing the rates of early marriage [14]. Other studies also support our findings, where teachers proved vital in making home visits to monitor students’ daily life activities, remained aware of their problems, and supported students’ education through parental involvement [40]. In addition, schools also engage students in extracurricular activities to fill their free time and help them grow their talents and interests beyond academia. Such strategies prevent students from entering into a relationship with the opposite sex and indirectly keep them away from early marriages and other sexual activities [40]. With the closure of schools during COVID-19, several critical services became inaccessible for adolescents that were previously provided by the schools, including school-based healthcare services and essential resources programs [44], which also contributed to early marriages and adolescent pregnancies. Besides the closure of schools, economic instability also promoted early marriages in low- and middle-income countries. The practice relieves the girl’s family from their economic burden in two ways: the prospect of receiving a dowry and the relief from having fewer family members to feed [45].

### 4.6. Preparedness for Future Crises

The review found that the COVID-19 pandemic created inconveniences and difficulties for adolescents in accessing SRH services. It was a struggle for most countries to be able to provide the needed SRH services for adolescents because of their unpreparedness for such global disasters. As mentioned by [22], countries must learn from the sudden interruption that the COVID-19 pandemic created in the healthcare delivery system. Hence, stakeholders, policymakers, and authorities must ensure the development of strategies that would help to mitigate such sudden difficulties when a disaster occurs [22]. In the United States of America, the pandemic allowed for the reassessment of the delivery of SRH services for hard-to-reach populations [46]. According to Hailemariam and colleagues [46], in situations where resources are limited among adolescents, strategies such as telehealth are impossible. Such inequities found in this current review included poor internet access and limited access to radio or televisions in less privileged communities and families. Although several strategies and interventions to prepare for future pandemics have been outlined and discussed by various authors [16,22,46], it is important to consider the context. An important situation that might be overlooked is the less privileged homes, located in the large cities. Although these families are situated in urban areas where the internet might be accessible and available, affordability might be a challenge. In such homes, the entire family might have only one television or radio, and with parents being at home during the lockdown, it is likely that the adolescent may not have the opportunity to watch the television or listen to the radio. Thus, using the media to address issues on SRH during a pandemic would not be beneficial to an adolescent in this class. These groups of adolescents are similar to those who are located in rural communities, and may not be considered for packages allotted to rural communities, because of their location. It is important to have such circumstances in mind when preparing for future pandemics. Adolescents in this class, including those in rural settings, might not own personal phones or have access to laptops or computers. Therefore, strategies or interventions requiring communication through social media platforms, text, or telephone discussion might not be feasible. The choices that would be made to prepare for the future must consider such important situations or barriers in order to make useful decisions.

Despite the preparedness to make SRH services available, accessible, and affordable to all in case of future pandemics, stakeholders and policymakers must consider the availability of resources and the culture of the people when adopting proposed strategies or programs. The review found that adolescents had access to SRH education and knowledge in schools and through their interactions with their teachers. In some countries or communities, introducing SRH services for adolescents in their homes could be a challenge. Discussion of topics pertaining to SRH has been mentioned to be restricted in some homes and communities [47,48] because it is unacceptable or frowned upon. Adolescents who are not exposed to the topic of SRH during this stage of their lives are likely to be naïve about the topic when they become adults. On the other hand, these adolescents might also learn about the topic from their peers or unauthorized social media platforms, [49,50] and could be at risk of acquiring incorrect information. Nonetheless, in countries where subjects on SRH are being taught at the basic and secondary level of education, adolescents are able to acquire the needed knowledge and services. With the COVID-19 lockdown in place, adolescents receiving education on SRH only from their schools had limited access to information. It is important to make provisions for such groups to have access to SRH services during future challenges. For example, future plans can be made where small outdoor group programs can be organized by the various schools in the communities on different days. This will allow adolescents to still have access to the needed SRH information and services without leaving their communities. Engaging the adolescents with recreational activities and social events in their communities would also keep them occupied and prevent unwanted pregnancies during future lockdowns [20]. Furthermore, a study in the United Kingdom found that service providers were hesitant to offer online consultancy and mentioned the need for the training of healthcare providers [51]. Hence, service providers must also be trained to be able to provide virtual consultations where needed.

### 4.7. Limitations and Future Research

There are limitations to this review that need to be highlighted. The findings of this scoping review may not be generalized to countries other than low- and middle-income countries. The findings cannot also be generalized to all LMICs because of the cultural and socioeconomic diversity amongst these countries. This scoping review also only included studies published in English; therefore, studies pertaining to this topic published in any other languages were excluded. Most qualitative studies included in this review were descriptive in nature; therefore, they did not provide an in-depth understanding of the findings. There were also limited numbers of published studies on this topic available.

The findings of the literature review on healthcare provide knowledge on the needed areas that require further research to improve the delivery of services. The gaps identified in the review show the need for further research, which will produce a wider understanding of the impact of a pandemic on the delivery of SRHS to adolescents and youth in low- and middle-income countries. One of the gaps identified in this review was that most of the studies focused on female adolescents. Therefore, further studies that target male adolescents are highly recommended. In addition, studies that would explore the multiple ways to ensure culturally accepted, reliable, available, and affordable SRH services for adolescents living in rural areas, adolescents with HIV, transgender adolescents, and refugees during future pandemics are highly encouraged. Moreover, the review found gaps in the use of other alternatives to provide SRH services during the lockdown. Hence, studies to improve the knowledge and ability of SRH service providers to use available technologies and other resources for the delivery of quality services when needed are also highly recommended.

## 5. Conclusions

The findings of this scoping review identified various aspects of adolescents’ sexual and reproductive health that were highly impacted due to the COVID-19 pandemic. These include limited access to sexual and reproductive health services, including access to contraceptives, menstrual products, and medications for HIV treatment; increased rates of early marriage due to school closures and a lack of support from school management; a rise in intimate partner and sexual violence; disruptions in maternity care; and increased involvement of adolescents in risky or exploitative work. Findings from the studies and policy papers used in this review revealed several interventions designed and suggested to address SRH and access to SRH services for adolescents in LMICs during and after the COVID-19 pandemic. These interventions include: using schools, mass media campaigns, and radio programs for comprehensive sex education, prevention of unwanted pregnancies, and generating awareness regarding hormonal therapies; involving healthcare professionals to ensure access to age-specific, accurate, and up-to-date information on SRH through the media and pamphlets; providing contraceptives for longer periods; improving access to contraceptives through alternative delivery modalities such as pharmacies and community-based delivery; utilizing telemedicine for providing easy access to services such as medical abortions, counselling, and screening to adolescents; using adolescent-friendly phone lines for advice; provision of HIV Self-Testing kits (HIVST) and home-based testing of other STIs; distribution of condoms, menstrual supplies, HIV pre-exposure prophylaxis medications (PrEP), lubricants, and douches; and utilizing digital platforms for improving the socialization of adolescents and providing psychosocial services. Efforts are required for developing socially and culturally appropriate policies by employing community-based models to support adolescents and lessen the impact of COVID-19 on them, hence reducing SRH issues among adolescents. Enhancing job opportunities and the provision of career counselling during such a pandemic would aid in eliminating social isolation and overcoming its negative effects on adolescents’ SRH. Lastly, only a few studies have been conducted in this area; hence, there is a dire need for further research to add to the current knowledge on the topic.

## Figures and Tables

**Figure 1 ijerph-18-13221-f001:**
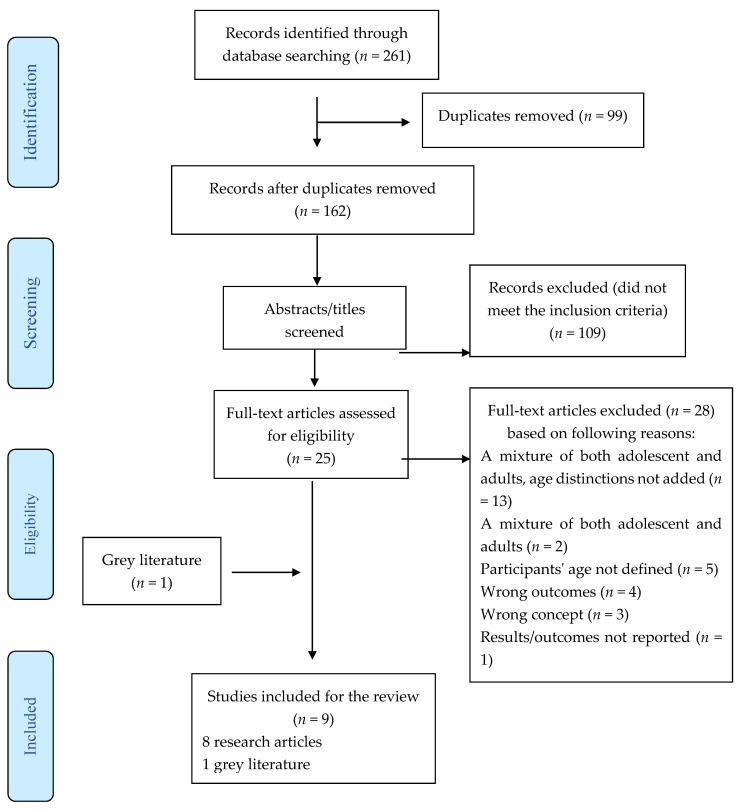
PRISMA flow diagram.

**Table 1 ijerph-18-13221-t001:** Characteristics of included studies (*n* = 9).

Author/s and Year of Publication	Setting and Country	Aim of Study	Study Design	Target Population	Intervention	Outcomes of Study
[14]	Settings: Urban, rural, and refugee camps. Multinational: Ethiopia, Cote d’Ivoire, and Lebanon	To mitigate the psycho-emotional toll of the pandemic and promote resilience among adolescents in some of the most difficult places in the world to be a young person.	Qualitative studies	Adolescents Age: 10–19 *n* = 568	No intervention	Exposure to early/child marriages (age 15) among some cultural groups, where the cost of marriage is cheaper. Pressure from parents on girls to marry because they were not in school due to the lockdown. Increase in intimate partner violence. Limited access to youth-friendly SRH care services. Negative coping strategies during the COVID-19 restrictions amongst some adolescents (suicidal thoughts, drinking of alcohol, smoking). Positive coping strategies during the restrictions (volunteering, gaming, sports, meditating, hobbies).
[15]	Settings: Urban Country: Kenya	To assess the psychosocial effects of COVID-19 within an ongoing cohort study of Adolescents Living with HIV.	Qualitative study	Adolescents and youth Age: 10–24 10–14 (*n* = 152), 15–19 (*n* = 188), 20–24 (*n* = 146)	No intervention	Some participants complained of not being able to go to their healthcare appointments. Some also reported not being able to have access to medication refills.
[16]	Country: Kenya	Three objectives: a. To document the likely challenges that can impede the implementation of a new media strategies model to disseminate SRHR messages to young people in media campaigns. b. To understand how media managers would mitigate the challenges that may interfere with using new media strategies model to disseminate SRHR messages in media campaigns to young people. c. To probe stakeholder perspectives on the possibility and acceptability of adapting a new media strategies model to pass SRHR messages to young people in media campaigns.	Qualitative study	Stakeholders (media managers, policymakers, healthcare professionals, teachers, parents, and youth) Age of youth = 15–24 Age of other participants = NR Total *n* = 54	The use of new media platform for sex education. Example Twitter, Facebook, WhatsApp. NGO created platforms for the youth.	Policymakers and decision-makers sometimes prevent the successful dissemination of SRHR messages. Churches and parents were opposition groups to media campaigns. Ignorance in the use of technology. Variation in the youth’s preference for media platforms could also interfere with the message.
[17]	Country: Jordan	To evaluate the impact of the lockdown circumstances in Jordan on antenatal care services and health circumstances of pregnant women during this period.	Qualitative study (cross-sectional study)	Pregnant women (15 years and above) Total *n* = 944 15–19 years (*n* = 4) 20–24 years (*n* = 146) 25 years and above (*n* = 794)	No intervention	Change in access to antenatal services since the beginning of the COVID-19 lockdown. Communication with care providers over the phone without actual antenatal visits; some had permits that allowed them to move freely during the curfew, and others used emergency medical services as an alternative to attending antenatal clinics. Some also reported no change in antenatal care schedule during the lockdown. Some pregnant women reported not being able to have access to antenatal visits during the lockdown. The majority of pregnant women who were not receiving antenatal services during the lockdown experienced pregnancy complications. These included gestational hypertensive disease, placenta previa, gestational diabetes, and vaginal bleeding.
[18]	Settings: Urban and rural Country: Kenya	To explore the pre-pandemic characteristics that may either protect girls from negative outcomes during the pandemic response or put them at higher risk.	Quantitative study	Adolescent girls Age: 10–14 years (*n* = 206) Age: 15–19 (*n* = 650)	No intervention	Sexual violence at the beginning of the COVID-19 pandemic. Some reported difficulty obtaining menstrual hygiene products. Most adolescent girls who were sexually active were 16 years and above.
[19]	Setting: Health center (urban) Country: South Africa	To assess the impact of the lockdown orders in response to the COVID-19 epidemic in South Africa on access to basic healthcare services.	Quantitative: Interrupted time series analysis	Adolescents 6–19 years of age (*n* = 4460) 20–45 years accounted for 48% (*n* = 22 231)	No intervention	Visits for perinatal care and family planning remained reasonably constant or modestly increased. Increased clinic visits for HIV immediately after the lockdown. This might have reflected an urgency to collect medications prior to an anticipated interruption in-clinic access or medication availability and/or national programmatic efforts to accelerate transitions to a new first-line regimen.
[20]	Country: Ecuador	To evaluate and learn from prior health policy in strategizing more effective adolescent pregnancy prevention legislation.	Descriptive paper (Policy analysis/discussion	Adolescents Age: 15–19 years *n* = NR	No intervention	Prevention of adolescent pregnancy. The new Model of Comprehensive Family, Community and Intercultural Health Care (MAIS-FCI) emphasized the equal level of care throughout each stage of an individual’s life cycle while also emphasizing that the adolescent population, in particular, continues to be intensely equipped with medical information. Promotion of health information as the main strategy, Intersectoral Policy for the Prevention of Pregnancy in Girls and Adolescents, which offers a plan that can serve as a starting point in the nation’s development of solutions that do not simply fault adolescents for adolescent pregnancy, but rather aim to lessen the social/cultural dynamics that act as an impetus.
[21]	Setting: UrbanCountry: Brazil	To describe the strategies adopted by PrEP1519 sites of Salvador and São Paulo to continue to provide HIV and SRH services during quarantine periods to contain the COVID-19 pandemic.	Follow-up program	Adolescent men sleeping with men (MSM) and transgender women (TGW) Age: 15–19. *n* = 484	Pre-exposure prophylaxis for 15–19 (PrEP1519) Program was being offered in the clinic and outreach. Social media and telemonitoring was set up to reach adolescent key population to continue the program and ensure access to SRH care and prevention needs.	The strategies intensified during the lockdown, as adolescents were eager to have a conversation and ask questions on how to persist with their PrEP use under quarantine. Adolescent key population used the platform to socialize and look for partners. Demand for the PrEP program services continued. Changes to online services were well received by participants and likely to continue after the quarantine.
[22]	Global	To continue responding to the sexual and reproductive health needs of adolescents in the context of the COVID-19 crisis.	Technical brief (grey literature)	Adolescents 10–19 years *n* = NR	SRHR policy and delivery improvement for adolescents’ access to SRH care services in LMICs	Policy implementation for: Provision of comprehensive sexuality education, contraceptive counselling and services, and comprehensive abortion care. Provision of antenatal, intrapartum, and postnatal care. Prevention and treatment of HIV and other STIs. Prevention, care, and response to sexual and gender-based violence. Counselling and services for sexual health and well-being, including the provision of menstrual health information and services.

## Data Availability

All data generated or analyzed during this study are included in this published article.

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
