# Peer review of "Impact of the COVID-19 Pandemic on Adolescents’ Sexual and Reproductive Health in Low- and Middle-Income Countries"

_ijerph, 2021, doi:10.3390/ijerph182413221_

Round 1

Reviewer 1 Report

Manuscript Number: ijerph-1458726

Title: Impact of the COVID-19 Pandemic on Adolescents’ Sexual and

Reproductive Health in Low- and Middle-Income Countries

The current article investigates the impact of the Covid-19 pandemic on the ability of adolescents who are living in low- and middle-income countries (LMIC) to access sexual and reproductive (SRH) health services. In particular, the article engages in a scoping review to identify and assess the extant literature concerning the impact of the Covid-19 pandemic on youth SRH needs and access to services for these populations.

As written, this manuscript currently possesses strengths as well as areas for improvement. The manuscript certainly has the potential to make an important contribution to the extant literature on both LMIC populations and research on access to SRH—especially given its focus on the ongoing context of a global, large-scale transformative event (i.e., Covid-19 pandemic). However, minor revision is required           before the reviewer can recommend publication. Please find reviewer comments below. 

Collectively, the Introduction/background sections provide a comprehensive review of the literature. The authors do a solid job of building the justification for the current study. Indeed, this is a timely and fruitful area of investigation. No revisions are recommended for this section. 

The Methods section is similarly comprehensive in its providing a detailed account of the authors’ strategy in conducting the scoping review of extant literature on the impact of the Covid-19 pandemic on SRH among adolescents who reside in LMICs and the types of interventions designed to improve their access to a range of SRH services. To be sure, a definite strength of this manuscript is the authors’ detailing of: Data sources and search strategy; study selection; and data extraction processes. The level of detail provided throughout this section—and each subsection—lends itself to trustworthiness.  

The Results section is strong but does require minor revision. For instance, at Line 132, page 3, the authors indicate the following: “The initial search retrieved a total of 261 articles. After removing duplicates and articles in other languages and reviewing abstracts with respect to the inclusion criteria, a total of twenty-five studies were considered relevant.” Presently, the manuscript discusses only inclusion criteria and does not describe or explain any exclusion criteria. This is standard in studies where the scoping review is the chosen method.  In order to recommend publication, this reviewer recommends that the authors sufficiently detail exclusion criteria alongside inclusion criteria. This is important given that the volume of articles initially retrieved for the scoping review was 261 and that subsequently only 25 were considered “relevant” and only 9 were included in for final review. Greater clarity around the process of narrowing down the literature is needed.  

Also, in the Results section, Line 148, page 4, in briefly discussing relevant themes, the authors state: “Adolescents faced various SRH vulnerabilities during Covid-19 which can be categorized into several themes including, limited access to sexual and reproductive health services, early marriages, sexual violence, disruption in maternity, etc.” This should be revised to include a list of all themes and should not abbreviated to, “etc.” More specifically, “increased involvement of adolescents in risky or exploitative work” should be listed here.

Additionally, in the Results section, the authors should confirm whether the following “themes” should be added to the abstract as neither theme is currently mentioned there, although they are described on pages 5 through 7: “Intervention to Improve Sexual and Reproductive Health Services During Covid-19” (see p. 5) and “Policy Development Related to Adolescent Sexual and Reproductive Health” (see pgs. 6-7).  In short, confirm the final list of themes and ensure these are consistent throughout the manuscript (Abstract, Introduction, Methods, Results, Discussion, and Conclusions). Please review and revise all relevant sections.

Finally, at a conceptual level, the authors provide little conceptualization or discussion of the term “low- and middle-income countries” (LMIC) anywhere in the manuscript. For example, on page 2, line 48, the authors begin by stating the following: “Globally, approximately 1.8 billion people are between the age of 10 and 19 years, and 90% of them live in low- and middle-income countries (LMICs).” The manuscript will be strengthened by a description of some universal characteristics of low- and middle-income countries. Broadly, what are some universal characteristics of LMICs? What specifically defines a low-income country? What defines a middle-income country?  What are the respective gross national incomes per capita within each category? How have other studies treated the category LMICs? These questions are provided not to be prescriptive but to provoke the authors to think more critically about the term “LMIC” and to give the reader a clear sense of why investigating the impact of Covid-19 on these particular contexts is crucial. In sum, this is a critical area of scholarship, particularly given its focus on the effects of the Covid-19 pandemic, and the authors should provide more definitional clarity around the term LMIC, given that it is a focal dimension of the study.

Contingent on the authors making the above revisions to the manuscript this reviewer believes that the manuscript will be suitable for publication in IJERPH.

Author Response

Response to Reviewer’s Comments

Reviewer One

  1. Collectively, the Introduction/background sections provide a comprehensive review of the literature. The authors do a solid job of building the justification for the current study. Indeed, this is a timely and fruitful area of investigation. No revisions are recommended for this section.

Authors Response: Thank you very reviewing our manuscript and also for the positive comments.

  1. The Methods section is similarly comprehensive in its providing a detailed account of the authors’ strategy in conducting the scoping review of extant literature on the impact of the Covid-19 pandemic on SRH among adolescents who reside in LMICs and the types of interventions designed to improve their access to a range of SRH services. To be sure, a definite strength of this manuscript is the authors’ detailing of: Data sources and search strategy; study selection; and data extraction processes. The level of detail provided throughout this section—and each subsection—lends itself to trustworthiness.

Authors Response: Thank you for the positive comments on our methods section.

  1. The Results section is strong but does require minor revision. For instance, at Line 132, page 3, the authors indicate the following: “The initial search retrieved a total of 261 articles. After removing duplicates and articles in other languages and reviewing abstracts with respect to the inclusion criteria, a total of twenty-five studies were considered relevant.” Presently, the manuscript discusses only inclusion criteria and does not describe or explain any exclusion criteria. This is standard in studies where the scoping review is the chosen method. In order to recommend publication, this reviewer recommends that the authors sufficiently detail exclusion criteria alongside inclusion criteria. This is important given that the volume of articles initially retrieved for the scoping review was 261 and that subsequently only 25 were considered “relevant” and only 9 were included in for final review. Greater clarity around the process of narrowing down the literature is needed.

Also, in the Results section, Line 148, page 4, in briefly discussing relevant themes, the authors state: “Adolescents faced various SRH vulnerabilities during Covid-19 which can be categorized into several themes including, limited access to sexual and reproductive health services, early marriages, sexual violence, disruption in maternity, etc.” This should be revised to include a list of all themes and should not abbreviated to, “etc.” More specifically, “increased involvement of adolescents in risky or exploitative work” should be listed here.

Additionally, in the Results section, the authors should confirm whether the following “themes” should be added to the abstract as neither theme is currently mentioned there, although they are described on pages 5 through 7: “Intervention to Improve Sexual and Reproductive Health Services During Covid-19” (see p. 5) and “Policy Development Related to Adolescent Sexual and Reproductive Health” (see pgs. 6-7). In short, confirm the final list of themes and ensure these are consistent throughout the manuscript (Abstract, Introduction, Methods, Results, Discussion, and Conclusions). Please review and revise all relevant sections.

Authors Response: We have included the exclusion criteria as recommended on page 3 line 124-128.

Articles that do not focus on adolescents in LMICs, access to SRH services during the pandemic, interventions implemented to promote SRH and access to SRH services for adolescents during were excluded. In addition, report that did not focus on SRH health experiences and/or outcomes of adolescents during the pandemic in LMICs were excluded from the review. Articles that focused on adolescent mental health were also excluded from this review.

We have included the other themes on line 24-29 page 1, and line 153-157 page 4 as recommended by reviewer.

  1. Finally, at a conceptual level, the authors provide little conceptualization or discussion of the term “low- and middle-income countries” (LMIC) anywhere in the manuscript. For example, on page 2, line 48, the authors begin by stating the following: “Globally, approximately 1.8 billion people are between the age of 10 and 19 years, and 90% of them live in low- and middle-income countries (LMICs).” The manuscript will be strengthened by a description of some universal characteristics of low- and middleincome countries. Broadly, what are some universal characteristics of LMICs? What specifically defines a low-income country? What defines a middle-income country? What are the respective gross national incomes per capita within each category? How have other studies treated the category LMICs? These questions are provided not to be prescriptive but to provoke the authors to think more critically about the term “LMIC” and to give the reader a clear sense of why investigating the impact of Covid-19 on these particular contexts is crucial. In sum, this is a critical area of scholarship, particularly given its focus on the effects of the Covid-19 pandemic, and the authors should provide more definitional clarity around the term LMIC, given that it is a focal dimension of the study.

Authors Response: We have inserted the recommended information in line 54-62 on page 2.

Low- and middle-income earning countries are countries that are considered as low-income earning countries and middle-income earning countries. According to World Bank low-income economies are countries with Gross National Income (GNI) per capita of $1,025 or less, and middle-income economies (low and high middle-income countries) are those with the GNI per capita between $1,026 and $12,375, using the World Bank Atlas method [3]. Thus, all countries with GNI per capita of $1,025 or less to $12,375 using the World Bank Atlas method can be classified as low-and middle-income earning countries.

Reference

  1. World Bank. Classifying countries by income. Available online: https://datatopics.worldbank.org/world-development-indicators/stories/the-classification-of-countries-by-income.html (assessed on 19 November 2021)

Contingent on the authors making the above revisions to the manuscript this reviewer believes that the manuscript will be suitable for publication in IJERPH

Reviewer 2 Report

The authors want to give a wide explaination of the state of art of accessing to Sexual and Reproductive Health Services during the COVID-19 pandemic in Low to Middle Income Countries. Despite the appropriate methodogy and discussion of the results, i have a few concerns about the manuscript:

  1. Reading the article title, it seems that you want to discuss about the impact of the COVID-19 Pandemic on Adolescents’ Sexual and Reproductive Health (SHR), though the manuscript exclusively focus on the issues about accessing to SRH services. For this reason, i strongly suggest to amend the title, including the "accessing to SHR services" reference.
  2. In the introduction section, you state that the reasons why adolescents and young adults access to SHR services are related with vulnerability to gender-based violence, early marriages, unintended pregnancy, and female genital mutilation, forgetting what mostly affect the sexual health: Sexual Dysfunctions.
    By including this reason, you could integrate the introduction section reporting a brief paragraph that explain the impact of COVID-19 pandemic on the sexual health of adolescents and adults. Here are a shortlist of useful articles to cite:
    • Grover S, et al. Sexual functioning during the lockdown period in India: An online survey. Indian J Psychiatry. 2021 Mar-Apr;63(2):134-141. doi: 10.4103/psychiatry.IndianJPsychiatry_860_20
    • Szuster E, et al. Mental and Sexual Health of Polish Women of Reproductive Age During the COVID-19 Pandemic - An Online Survey. Sex Med. 2021 Aug;9(4):100367. doi: 10.1016/j.esxm.2021.100367.
    • Sansone A, et al. "Mask up to keep it up": Preliminary evidence of the association between erectile dysfunction and COVID-19. Andrology. 2021 Jul;9(4):1053-1059. doi: 10.1111/andr.13003.
    • Omar SS, et al. Psychological and Sexual Health During the COVID-19 Pandemic in Egypt: Are Women Suffering More? Sex Med. 2021 Feb;9(1):100295. doi: 10.1016/j.esxm.2020.100295.
    • Mollaioli D, et al. Benefits of Sexual Activity on Psychological, Relational, and Sexual Health During the COVID-19 Breakout. J Sex Med. 2021 Jan;18(1):35-49. doi: 10.1016/j.jsxm.2020.10.008. 

Author Response

Response to reviewer’s comments

Reviewer Two

  1. Reading the article title, it seems that you want to discuss about the impact of the COVID-19 Pandemic on Adolescents’ Sexual and Reproductive Health (SHR), though the manuscript exclusively focus on the issues about accessing to SRH services. For this reason, I strongly suggest to amend the title, including the "accessing to SHR services" reference.

Authors Response: Thank you for your recommendation to amend the title. We the authors believe that because there are currently limited articles and reports on the topic, it is not a good idea to narrow it to only access to SRH services in LMIC during the pandemic. We will want to leave it as a broader topic as it is now, although we agree with you that majority of the articles and report we found focused on access to SRH services during the pandemic. 

  1. In the introduction section, you state that the reasons why adolescents and young adults access to SHR services are related with vulnerability to gender-based violence, early marriages, unintended pregnancy, and female genital mutilation, forgetting what mostly affect the sexual health: Sexual Dysfunctions.
    By including this reason, you could integrate the introduction section reporting a brief paragraph that explain the impact of COVID-19 pandemic on the sexual health of adolescents and adults. Here are a shortlist of useful articles to cite:
    • Grover S, et al. Sexual functioning during the lockdown period in India: An online survey. Indian J Psychiatry. 2021 Mar-Apr;63(2):134-141. doi: 10.4103/psychiatry.IndianJPsychiatry_860_20
    • Szuster E, et al. Mental and Sexual Health of Polish Women of Reproductive Age During the COVID-19 Pandemic - An Online Survey. Sex Med. 2021 Aug;9(4):100367. doi: 10.1016/j.esxm.2021.100367.
    • Sansone A, et al. "Mask up to keep it up": Preliminary evidence of the association between erectile dysfunction and COVID-19. Andrology. 2021 Jul;9(4):1053-1059. doi: 10.1111/andr.13003.
    • Omar SS, et al. Psychological and Sexual Health During the COVID-19 Pandemic in Egypt: Are Women Suffering More? Sex Med. 2021 Feb;9(1):100295. doi: 10.1016/j.esxm.2020.100295.
    • Mollaioli D, et al. Benefits of Sexual Activity on Psychological, Relational, and Sexual Health During the COVID-19 Breakout. J Sex 

Authors Response: Thank you once again for this comment and also providing us with the references. We the authors do not think it is important to have a separate section in the introduction that focuses on sexual dysfunction. Our paper is reviewing the impact of COVID-19 on adolescent SRH in LMIC, and we prefer to remain focus within this broader topic and not narrow down to a specific health issue.   

Reviewer 3 Report

Overall, I believe that it may be prudent to delay publication of the review since the evidence that the analysis is based on is too weak to make an informed decision. There are few publications used within the scoping review and at times, it appears that decisions are made based on only one article. Many of the articles are of low quality suggesting that the evidence is not rigorous and this may lead to misleading policy and program directions in terms of the needs of adolescents in low and middle income countries. 

I agree that there is a paucity of evidence on the SRH needs of adolescents in low and middle income countries.

There is a problematic with the format when citing within text.  This is repeated throughout the document and must be corrected. For example, page 2, line 84, is written as "...methodology as guided by [9] [10]. "  This should be corrected to  "... methodology as guided by [author] (ref) and [author] (ref).

Other specific  comments are as follows:

pg 1, line 17 Why state 'already'?

pg 1. line 22. What is the rationale for removing full text? I would expect full text to be kept.

pg1. Line 36 in 'a' century should read ...in 'this, century

pg 2. line 64. The thesis statement is weak so the follow up sentences are not strongly supported.

pg 2. line 69. Review was to identify and assess.... perhaps if the review was to summarize the existing literature, I would be more convinced of its worth.

pg 2. line 79. I believe that this type of review over reaches its potential. The literature is not mature to rigorously identify the gaps/needs.

pg 2. line 96 MK is not listed as an author.

page 3 follows:

search strategy should include examples of the search terms, time or period when the search was done.

There is also some contradictions on the type of restrictions done.

line 145. It is very odd to reference one technical report with two references  (...one [22] technical report [23]). 

There is a problem with lines 159 and 174 (did they have access or not to FP services?)

line 160. Do not start a sentence with Besides

line 162. The material after [14] is different from the above and should be a new paragraph. Try to integrate the above sentence elsewhere.

line 183. explain what is level 5 lockdown

lines 187-196. The evidence is very weak. Was this similar for boys and girls? 

Some of these adolescents are living in conflict zones. This is a special circumstance that could not be applied to those living in stable societies.

Page 5

line 203. Do you mean incidence as in the epidemiological sense. If not, you may consider using another word. 

line 208. This is a difficult claim since there is no direct comparison to the precovid time. The last 3 lines of the paragraph states the opposite of the first sentence.

line 219. Is Afar being referred to as a country?

In essence, there is a need for a thorough review of the article.

Author Response

Response to reviewer’s comments

Reviewer 3

  1. Overall, I believe that it may be prudent to delay publication of the review since the evidence that the analysis is based on is too weak to make an informed decision. There are few publications used within the scoping review and at times, it appears that decisions are made based on only one article. Many of the articles are of low quality suggesting that the evidence is not rigorous and this may lead to misleading policy and program directions in terms of the needs of adolescents in low and middle income countries. 

I agree that there is a paucity of evidence on the SRH needs of adolescents in low and middle income countries.

Authors Response: Thank you for your comment. The authors believe that the shorter time frame of articles included in our review contributed to the reason for the perception of low-quality article you have highlighted. However, the findings of this review contain the information needed to implement policies on SRH services during and after a pandemic.

There is a problematic with the format when citing within text.  This is repeated throughout the document and must be corrected. For example, page 2, line 84, is written as "...methodology as guided by [9] [10]. "  This should be corrected to  "... methodology as guided by [author] (ref) and [author] (ref).

Authors Response: Thank you for your feedback. We have corrected this in the manuscript.

Other specific comments are as follows:

  1. pg 1, line 17 Why state 'already'?

pg 1. line 22. What is the rationale for removing full text? I would expect full text to be kept.

Authors Response: We have corrected this on line 22&23 page 2. It is not removing full-text but rather full-text reading of all articles.

  1. Line 36 in 'a' century should read ...in 'this, century

Authors Response: Corrected on line 41

  1. pg 2. line 64. The thesis statement is weak so the follow up sentences are not strongly supported.

Authors Response: Changes made on line 77-78

  1. pg 2. line 69. Review was to identify and assess.... perhaps if the review was to summarize the existing literature, I would be more convinced of its worth.

Authors Response: Corrected on line 82. We have now made it the review was to identify and summarize.

  1. pg 2. line 79. I believe that this type of review over reaches its potential. The literature is not mature to rigorously identify the gaps/needs.

Authors Response: Literature included articles were not strong enough but however we believe this literature will help identify the gaps.

  1. pg 2. line 96 MK is not listed as an author.

Authors Response: MK has now been added to the list of Authors.

  1. page 3 follows:

search strategy should include examples of the search terms, time or period when the search was done.

There are also some contradictions on the type of restrictions done.

Authors Response: Search term used for each data are included in the supplementary document 1 attached to the manuscript. We have listed few of the search terms we used on line 118-122 on page 3. The month and year when the search was done has also been stated on page 3 line 126-127. We have not applied any restrictions to our search and this has been stated clearly on line 122-123, page 3

  1. line 145. It is very odd to reference one technical report with two references  (...one [22] technical report [23]). 

Authors Response: This has been corrected we have only one refence for that. Page 4 line 169

  1. There is a problem with lines 159 and 174 (did they have access or not to FP services?)

Authors Response: We have made the changes to these statement on line 202-212 page 5.

  1. line 160. Do not start a sentence with Besides

Authors Response: We have made changes to this statement on line 188 page 4

  1. line 162. The material after [14] is different from the above and should be a new paragraph. Try to integrate the above sentence elsewhere.

Authors Response: The entire material has been but into one paragraph> on Page 4 line 189-194.

  1. line 183. explain what is level 5 lockdown

Authors Response: The level five lock down has been explained on page 5 line 214-215.

  1. lines 187-196. The evidence is very weak. Was this similar for boys and girls? 

Authors Response: This is the evidence we found in the literature, and the difference between the situation amongst adolescent boys and the girls has been clearly stated based on what was reported in the articles.

  1. Some of these adolescents are living in conflict zones. This is a special circumstance that could not be applied to those living in stable societies.

Authors Response: Thank you for the comments. We agree but for this review we have not included any study conducted in conflict zones and this was not the objective of this review.

  1. Page 5 line 203. Do you mean incidence as in the epidemiological sense. If not, you may consider using another word. 

Authors Response: We have changed the word from incidence to frequency. On page 5 line 235.

  1. line 208. This is a difficult claim since there is no direct comparison to the precovid time. The last 3 lines of the paragraph states the opposite of the first sentence.

Authors Response: We have clarified this on page 5 line 245.

  1. line 219. Is Afar being referred to as a country?

Authors Response: Afar is a town in Ethiopia. We have clarified this on page 6 line 252

  1. In essence, there is a need for a thorough review of the article.

Authors Response: We have thoroughly reviewed our manuscript. Thank you for your feedbacks and suggestions.

Round 2

Reviewer 3 Report

Impact of the COVID-19 Pandemic on Adolescents’ Sexual and Reproductive Health in Low- and Middle-Income Countries

This is a topic of interest to the journal and in my last review, I suggested various revisions to the authors.  While some revisions were made, I am disappointed that the authors did not take the time to review all of my overall comments. As such, I continued to see the same errors which I had previously noted.

General comments

It cannot be over stressed that the article should undergo an in-depth review, making corrections even in areas where the reviewers have erred to suggest. There are still grammatical errors such as those found on line 143 – 148 where there is a change in tense within the same paragraph.

…Articles that do not…..Reports that did not

There is missing information.

I could not find Table 1 or Figure 1.

Finally, pay attention to references to ensure their accuracy.

Specific comments (per line number):

171-177

  • Colon rather than a comma is needed after ‘including’.

The word ‘including’ suggests that other themes are presented but no others were documented in the RESULTS section

  • Themes 1 (limited access to SRH) to 5 (involvement in risky or sexually exploitative work) are SRH vulnerabilities. However, themes 6 and 7 (interventions and policy development) should not be included under SRH vulnerabilities since they are awkward and may need another subtopic.
  • 178 Findings of these issues are discussed… (Present tense should be used)
  •  

197 Please reorganize the sentence. Perhaps review “IN”?

281, 288, 337, 339, 341, etc. E.g. the [23] …. Please correct these errors throughout the text.

317 Please review the sentence. Perhaps lines 318 should be part of the previous sentence?

321 ‘assessed’ rather than ‘accessed’?

332 Consider reorganizing the sentence.

377 – ‘ALSO’ suggests that you have already discussed the factors contributing to SRH during lockdown in this paragraph or the previous paragraph. However, this is the first time it is being mentioned in the discussion section. As such, consider revising the statement.

378 ‘Found’ may not be the most appropriate word as ‘found’ suggests that the information came from rigorous research. However, this is a scoping review and assessing the studies for ‘rigor’ may not have been part of the process. As such, I would suggest to use the word ‘suggest’ instead of ‘found’.

393, 408 ditto (found)

395-97 Consider reorganizing the sentence

430-32 Canadian data is cited in reference 31/33. I could not find any studies relating to Canada in these references. Kindly review

466-469 Although I do not disagree with the statement, I could not find any information in the results of your study to suggest that teachers play a protective role in reducing early marriage, etc. As such, the information appears as an overreach.

503 Media is discussed as involving TV only. Kindly revise.

541 I would suggest removing the word “few”. There are many limitations to the study as 1. You rightly suggested the limited number of studies available. 2. These studies did not fully encompass the diversity of LMICs. LMICs are most have a singular commonality of having a GNI or$3995 or less. There is no cultural or social ties between these countries – factors which are important determinants of health. The findings cannot be generalized to all LMICs due to this factor.

566-568 No new information should be presented here. The information presented in the results did not including smoking or alcohol abuse. In fact, the information presented was not of alcohol use/abuse by youth but youth involvement in the trade of alcoholic beverages.

576-580 Are these suggestions or are these findings from the studies/policy papers on COVID used in this scoping review? Consider this query and revise as appropriate.

Author Response

Reviewer 3 Round 2

This is a topic of interest to the journal and in my last review, I suggested various revisions to the authors.  While some revisions were made, I am disappointed that the authors did not take the time to review all of my overall comments. As such, I continued to see the same errors which I had previously noted.

Authors Responds: Thank you for your comments and suggestions, we have responded to your comments to our manuscript, and highlighted them below.

Reviewers Comments

General comments

It cannot be over stressed that the article should undergo an in-depth review, making corrections even in areas where the reviewers have erred to suggest. There are still grammatical errors such as those found on line 143 – 148 where there is a change in tense within the same paragraph.

…Articles that do not…..Reports that did not

Authors Responds: We have looked at these grammatically errors and have made the needed corrections throughout the manuscript.

Reviewers Comments

There is missing information.

I could not find Table 1 or Figure 1.

Authors Responds: We included the table 1 and figure 1 when we initially submitted the manuscripts to the journal. However, we have included these two supplementary documents at the bottom of the manuscript as requested.

Reviewers Comments

Finally, pay attention to references to ensure their accuracy.

Authors Responds: We have revised all refences particularly in text referencing as mentioned.

 Reviewers Comments

Specific comments (per line number):

171-177

  • Colon rather than a comma is needed after ‘including’.

Authors Responds: We have changed the comma to a colon on line 173 page 4.

The word ‘including’ suggests that other themes are presented but no others were documented in the RESULTS section

  • Themes 1 (limited access to SRH) to 5 (involvement in risky or sexually exploitative work) are SRH vulnerabilities. However, themes 6 and 7 (interventions and policy development) should not be included under SRH vulnerabilities since they are awkward and may need another subtopic.
  • 178 Findings of these issues are discussed… (Present tense should be used)
  •  

197 Please reorganize the sentence. Perhaps review “IN”?

Authors Responds: These corrections have been made on the specific lines and pages.

281, 288, 337, 339, 341, etc. E.g. the [23] …. Please correct these errors throughout the text.

Authors Responds: We have corrected these errors in the manuscripts on the listed lines and pages, and all other pages with similar errors.

317 Please review the sentence. Perhaps lines 318 should be part of the previous sentence?

Authors Responds: We have made the correction on line 317 and 318 on page 7.

321 ‘assessed’ rather than ‘accessed’?

Authors Responds: We have made this correction in the manuscript on this line 321 page 7.

332 Consider reorganizing the sentence.

Authors Responds: The sentence has been re organised on line 332 page 7.

377 – ‘ALSO’ suggests that you have already discussed the factors contributing to SRH during lockdown in this paragraph or the previous paragraph. However, this is the first time it is being mentioned in the discussion section. As such, consider revising the statement.

Authors Responds: We have removed the word also and corrected this statement.

378 ‘Found’ may not be the most appropriate word as ‘found’ suggests that the information came from rigorous research. However, this is a scoping review and assessing the studies for ‘rigor’ may not have been part of the process. As such, I would suggest to use the word ‘suggest’ instead of ‘found’.

393, 408 ditto (found)

Authors Responds: We have changed the word found to suggest on line 378, 393 and 408.

395-97 Consider reorganizing the sentence

Authors Responds: The sentence on line 395-397 has been re structured.

430-32 Canadian data is cited in reference 31/33. I could not find any studies relating to Canada in these references. Kindly review

Authors Responds: We have revised the information in the reference 31-32 on line 430-432 on page 9.

466-469 Although I do not disagree with the statement, I could not find any information in the results of your study to suggest that teachers play a protective role in reducing early marriage, etc. As such, the information appears as an overreach.

Authors Responds: We the authors believe that we need to leave this statement in the discussion as it is, because on line 230-231 page 5 states “Even daughters in their early adolescence were forced to get married because of the limited presence of local authority officials and schoolteachers who otherwise would cause hindrance in early child marriages”. This was from reference #15.

503 Media is discussed as involving TV only. Kindly revise.

Authors Responds: We have revised this statement and included the radio on line 504. Also reading further in that paragraph on line 506-509 page 11 mentioned other media platforms.

541 I would suggest removing the word “few”. There are many limitations to the study as 1. You rightly suggested the limited number of studies available. 2. These studies did not fully encompass the diversity of LMICs. LMICs are most have a singular commonality of having a GNI or$3995 or less. There is no cultural or social ties between these countries – factors which are important determinants of health. The findings cannot be generalized to all LMICs due to this factor.

Authors Responds: We have corrected this on page 541 and included your suggestions on line 545-550 page 11.

566-568 No new information should be presented here. The information presented in the results did not including smoking or alcohol abuse. In fact, the information presented was not of alcohol use/abuse by youth but youth involvement in the trade of alcoholic beverages.

Authors Responds: We have corrected this error and deleted the statement on smoking and alcohol abuse on line 571 page 12.

576-580 Are these suggestions or are these findings from the studies/policy papers on COVID used in this scoping review? Consider this query and revise as appropriate.

Authors Responds: We have revised this statement and made a statement prior to these suggesting on line 572 page 12

Round 3

Reviewer 3 Report

Thank you for your quick response.